

# Potential impacts of chemical weathering on feldspar luminescence dating properties

Melanie Bartz[1], Jasquelin Peña[2], Stéphanie Grand[1], Georgina E. King[1]

[1]Institute of Earth Surface Dynamics, University of Lausanne, Lausanne, 1015, Switzerland
[2]Department of Civil and Environmental Engineering, University of California, 760 Davis Hall, Berkeley, USA

*Correspondence to*: Melanie Bartz (melanie.bartz@unil.ch)

**Abstract.** Chemical weathering alters the chemical composition of mineral grains. As a result, trapped-charge dating signals of primary silicates may be progressively modified. In this study, we artificially weathered three feldspar specimens to
understand the effect of proton- and ligand-promoted dissolution on their luminescence properties. We conducted kinetic experiments over 720 h using two solutions: (1) oxalic acid (pH 3, 20 °C), an organic acid with chelating abilities, and (2) aqua regia (pH <1, 40 °C), a mixture of strong acids creating aggressive acid hydrolysis conditions. These two solutions were chosen to approach over laboratory timescales some of the changes that may occur over geological time scales as minerals weather in nature.

The effect of the extracting solutions on mineral dissolution was investigated by following the concentration of elements accumulating in solution, while changes in feldspar surface morphology was assessed by scanning electron microscopy (SEM). Subsequent changes in feldspar luminescence in the near-UV (~340 nm) and blue (~410 nm) thermoluminescence (TL) and infrared stimulated luminescence (IRSL) emission bands were assessed at the multi- and/or single-grain levels to gain insight into the emission spectra, dose response, saturation, and anomalous fading characteristics of the feldspars. In all experiments,
only minor feldspar dissolution was observed after 720 h with <5 % of total Al, Si, Na, and Ca appearing in the aqueous phase, while 5-8 % of the total Mn and Fe were extracted. In general, aqua regia, the more chemically-aggressive solution, had a larger effect on feldspar dissolution compared to that of oxalic acid. Additionally, our results showed that although the TL and IRSL intensities changed slightly with increasing artificial weathering time, the feldspar luminescence properties were otherwise unmodified. This suggests that chemical alteration of feldspar surfaces may not affect luminescence dating signals
obtained from natural samples.

## 1 Introduction

Trapped charge dating monitors the ionising radiation induced population of defects by electrons and holes. Specifically, exposure to heat or light causes the electron population to be evicted, resulting in the emission of light and forming the basis for luminescence dating (Aitken, 1985; Huntley et al., 1985). Luminescence dating can be used to constrain the rock's or
sediment's last exposure to heat (e.g., Aitken, 1985), or light (e.g., Huntley et al., 1985), or the cooling history of bedrock





(e.g., Guralnik et al., 2015; King et al., 2016). Feldspar minerals constitute almost half of the oceanic and continental crusts (Parsons, 2010) and even the purest feldspar crystal contains a vast number of defects (e.g., impurities). These defects are sensitive to the effects of ionising radiation (Aitken, 1985; Marfunin, 1979). Alteration of feldspars due to chemical weathering occurs ubiquitously on the Earth's surface from shallow to deeply buried rocks (Yuan et al., 2019) and may modify the

chemical composition of mineral grains (e.g., leaching of elements, Nesbitt et al., 1980). Thus, it follows that luminescence signals may also be modified.

Previous studies have observed that chemical weathering influences the luminescence properties of silicate minerals (Jeong et al., 2007; Jeong and Choi, 2012), and such changes have been used as indicators of relative weathering extent (Wang and Miao, 2006). Additionally, it has been proposed that stratigraphic age reversals in loess-palaeosol sequences, assessed through

luminescence methods, may be attributed to the highly weathered nature of the sediments (Berger et al., 2001) and that variability in anomalous fading rates in bedrock feldspars might be correlated with weathering processes (Huntley, 2011; Valla et al., 2016). However, the effect of chemical weathering on luminescence properties has not been studied systematically. Without a better understanding of the effect of weathering on feldspar luminescence signals, luminescence chronologies over Quaternary timescales might be biased. Thus, two main research questions are targeted in this study: (i) does chemical

weathering change the luminescence properties of feldspar, and (ii) what are the impacts of these changes on luminescence dating? To answer these questions, we studied the effect of partial dissolution on the luminescence properties of feldspar minerals important for luminescence dating. We artificially weathered K- and Na-feldspars in time-series experiments up to 720 h using acidic solutions. Luminescence analyses at the multi- and/or single-grain levels were carried out to gain insights into thermoluminescence (TL) emissions, infrared stimulated luminescence (IRSL) dose response, saturation, and anomalous

fading characteristics of the feldspars, essential parameters in luminescence dating applications (e.g., Buylaert et al., 2012; Thomsen et al., 2008).

## 2 Feldspar crystal structure, luminescence centres, and their relationship to chemical weathering

Feldspars belong to the group of aluminosilicates having a general structure of $MT_4O_8$, where M is usually $Ca^{2+}$, $Na^+$ or $K^+$, and T is $Al^{3+}$ or $Si^{4+}$ (Ribbe, 1983). Feldspars exhibit partial ternary solid solutions between three pure components, namely

$CaAl_2Si_2O_8$ (anorthite), $NaAlSi_3O_8$ (albite), and $KAlSi_3O_8$ (orthoclase), forming the alkali feldspar series between the K- and Na-end-members and the plagioclase feldspar series between the Na- and Ca-end-members (Deer et al., 2013; Parsons, 2010). Depending on crystallisation temperature and thermal history, feldspars can be structurally classified as ordered or disordered based on the distribution of $Al^{3+}$, substituted for $Si^{4+}$ (Ribbe, 1983). Various elements can substitute for K, Na, Ca, Al, and Si in the crystal lattice due to ionic characteristics, structural aspects, or formation processes, acting in some cases as luminescence

centres (Krbetschek et al., 1997).

Most feldspars show blue emission centred at ~410 nm (see overview in Riedesel, 2020). The blue emission has been associated with a hole centre located on Al-bridging O ions (Finch and Klein, 1999; Speit and Lehmann, 1982), to $Eu^{2+}$ substituting for



$Ca^{2+}$ on M-sites (Götze et al., 1999; Krbetschek et al., 1997), or to $Ti^{4+}$ activation (Mariano and Ring, 1975). Recently, Riedesel et al. (2021) came to the conclusion that Al-O-Al bridges are related to the blue luminescence emission as a function of Al-Si

disorder, which is in agreement with previous observations (e.g., Speit and Lehmann, 1982). Whilst the blue emission is of particular interest for luminescence dating due to suitable luminescence dating properties (e.g., Buylaert et al., 2012), the isolation of the blue emission signal is challenging due to broad emission detection bands during luminescence measurements. For example, overlap with the blue luminescence emission and the near-UV (~300-360 nm) and green-yellow emission (~550-580 nm) bands has been found in various feldspar minerals (see Riedesel, 2020 for a review). Whilst the UV emission has

been related to an intrinsic defect (Garcia-Guinea et al., 1999), the green-yellow emission is often ascribed to $Mn^{2+}$ in Ca-sites (Geake et al., 1977), although this has recently been questioned by Prasad et al. (2016).

Chemical weathering of feldspars leads to the breaking of bonds and the release of interstitial and framework elements to solution as well as changes in surface morphology and chemistry (Chardon et al., 2006; Yuan et al., 2019). Thus, chemical and crystallographic changes during weathering might result in changes in the defects responsible for luminescence production.

In particular, one may ask where luminescence signals in the mineral grain originate from. In luminescence dating, it is generally assumed that luminescence originates from the entire grain rather than the surface, although light emitted from the centre of the grain would be re-absorbed by the grain itself (Duller, 1997). Additionally, optical absorption properties are different for different members of the feldspar groups and depend on grain properties such as transparency and grain size. Duller (1997) noted that pure feldspars absorb weakly in the visible part of the spectrum, but an absorption peak has been

observed in the UV part of the spectrum at ~320 nm. Since pure feldspar rarely exist in nature, trace and major elements, cracks, inclusions, and exsolution phenomena can result in a variety of mineral colours, which limit optical transmission (Duller, 1997; Hofmeister and Rossman, 1983). As chemical weathering is mostly a surface phenomenon (Yuan et al., 2019), we hypothesize that the luminescence emitted from feldspars with limited optical transmission will be influenced by chemical weathering.

**3 Material and methods**

All chemical, mineralogical, and luminescence experiments were carried out at the University of Lausanne (Switzerland).

**3.1 Samples**

Feldspar specimens from the Natural History Museum at the University of Oslo (Norway) were used in this study. These include samples ALB1 (KNR32491) and ALB2 (KNR16962), which are two plagioclase samples close to the albite end-

member, and sample MIC (KNR32141), which is a microcline K-feldspar. In order to compare our results to luminescence studies of sediment dating, the samples were ground manually with an agate mortar and pestle and sieved to a grain size fraction of 150-212 μm. Fines were removed by washing the powder in an ultrasonic basin with distilled water until the supernatant remained clear. The samples were then dried in an oven at 40 °C.



The chemical composition and mineralogy of the samples were determined by X-ray fluorescence (XRF) and X-ray diffraction
(XRD), respectively. For both analyses, the 150-212 µm fraction was manually ground to a fine powder (~20 µm). XRF
measurements of major elements were carried out on fused glass discs on a Phillips PW 2400 spectrometer equipped with a
rhodium tube, following the methods described by Pfeifer et al. (1991). XRD analyses were conducted using a Thermo ARL
X'TRA Powder Diffractometer. Approximately 100 mg of powder sample was analysed using Cu Kα radiation covering the
1-65° 2θ range according to Adatte et al. (1996) and compared to the RRUFFTM Project mineralogical database
(https://rruff.info/).

### 3.2 Chemical alteration experiments

We conducted two artificial weathering experiments using each of the three feldspar samples. Experiments were performed in
batch mode using two different solutions with five sampling times: 0 (unweathered), 4, 96, 240, and 720 h. After each time
point, the solids and filtrates were kept for further analyses.

In the first set of experiments, samples were treated with an aqua regia solution (3:1 mixture of hydrochloric and nitric acids;
USEPA 3050 and ISO standard 11466). Teflon containers were filled with 0.25 g of sample material and 2.5 ml of solution.
The pH of this solution was <1. The containers were closed and placed in an oven at a constant temperature of 40 °C, and
shaken over the specific time periods with a shaking incubator. The aqua regia method was chosen because it induces strong
acid hydrolysis conditions, which allow it to efficiently leach transition metals and trace elements from the surface of minerals,
without destructuring the silicate framework (and thus completely dissolving the mineral).

In the second set of experiments, samples were treated with a 0.2 M oxalic acid – oxalate buffer (Schwertmann, 1973). The
pH of this solution was $3.0 \pm 0.1$. The extraction was conducted in the dark to prevent Fe(III) reduction. Falcon centrifuge
tubes (50 ml) were filled with 0.25 g sample material and 12.5 ml of solution. The extraction solution was prepared by
dissolving 1.62 g of ammonium oxalate monohydrate $(NH_4)_2C_2O_4.H_2O$ and 1.08 g of oxalic acid hydrate $COOH_2.2H_2O$ in 100
ml of deionized water. The tubes were shaken on a mechanical shaker table at 175 revolutions per minute at room temperature.
This organic extractant was chosen because it offers weak acid hydrolysis conditions coupled with chelating ability of $Al^{3+}$
and $Fe^{3+}$. The oxalate extraction is known to complex and release Al, Fe and Si from minerals having low crystalline order,
and mimics a naturally-occurring chemical weathering pathway. Plants have been shown to exude oxalic acid to promote
mineral weathering and acquire essential nutrients (Collignon et al., 2012; Keiluweit et al., 2015).

Total dissolved concentrations of Si, Al, K, Ca, Na, Mg, Mn, and Fe in the solutions after each weathering time point were
analysed by inductively coupled plasma-optical emission spectroscopy (ICP-OES) using an Perkin Elma Optima 8300. For
each sample and each element, three emission lines were measured in triplicate. Finally, Sc was used as an internal standard.
Congruent and incongruent dissolution of samples was tested using the mole ratio of Al to Si in the solids and in the solutions.



### 3.3 Grain surface morphology

Images of feldspar grains of samples ALB1, ALB2, and MIC were obtained using a Tescan Mira II LMU scanning electron microscope (SEM) operated in high vacuum. Energy dispersive X-ray (EDX) analyses were performed using a Penta-FET 3x detector monitored by the AZtec 2.4 software package released by Oxford Instruments. Twenty grains of unreacted and artificially weathered grains were analysed in order to visualize the impact of aqua regia and oxalic acid on the grain surface.

### 3.4 Luminescence analyses

**3.4.1 Instrumentation**

Multi-grain luminescence measurements used an automated Risø TL/OSL-DA-20 reader equipped with a Detection And Stimulation Head (DASH) system (Lapp et al., 2015). The DASH system had a $^{90}Sr/^{90}Y$ beta source delivering a dose rate of ~0.15 Gy/s and infrared (IR, ~850 nm) light emitting diodes (LEDs). TL and IRSL signals were detected using an Electron Tube PDM 9107Q-AP-TTL-03 (160-630 nm) attached to the DASH. In addition, TL emission spectra were measured using

an Andor Kymera 193i spectrograph (wavelength and efficiency calibrated) equipped with an Andor iXon Ultra 888 electron multiplying charge coupling device (EMCCD, 300-900 nm) camera attached to the DASH system. The EMCCD has a 1024 x 1024 sensor format and a pixel size of 13 µm. The spectral resolution of the system is 13 nm for all spectra recorded. Single-grain luminescence measurements used an automated Risø TL/OSL-DA-20 reader equipped with a $^{90}Sr/^{90}Y$ beta source for irradiation delivering a dose rate of ~0.08 Gy/s. A single-grain attachment including an IR laser (~830 nm) were used (Bøtter-

Jensen et al., 2003). We used a 7.5 mm Hoya U-340 filter, and a blue filter pack (Schott BG3/Schott BG39) to measure the luminescence in the UV (~340 nm), and blue (~410 nm) emission bands, respectively.

### 3.4.2 Thermoluminescence (TL) spectral measurements

Three multi-grain aliquots per time point were mounted on stainless steel discs (6 mm); the mass of the grains were weighted for TL spectroscopy. TL signals were recorded from each disc without any optical filters. The vertical shift speed and horizontal

shift readout speed were set to 4.33 µs and 30 MHz, respectively. The spectra consist of 1024 bins with a bin width of 0.43 nm between 281 and 717 nm (centred at 500 nm). The grating of the spectrograph was set to 150 lines/mm and 500 nm blaze. The aliquots were given a beta dose of 270 Gy and TL measurements were made by heating the samples to 400 °C at a rate of 1 °C/s, detecting the emitted light as a function of wavelength. After measurement of the TL emissions, the same discs were measured to correct for background noise. The resulting TL spectra were corrected for sample mass (~9 mg) and electron-

multiplying (EM) gain to allow absolute signal intensity comparisons.

### 3.4.3 Infrared stimulated luminescence (IRSL) dose response and fading measurements

Three multi-grain aliquots (2 mm) were measured for each time point of the weathering experiments with an elevated temperature post infrared (pIR)-IRSL single-aliquot regenerative-dose (SAR) protocol (Buylaert et al., 2009; Thomsen et al.,

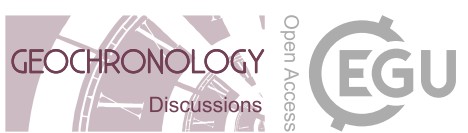

2008) using a preheat of 250 °C (for 60 s, 1 °C/s), a first IR stimulation temperature of 50 °C (IR$_{50}$ for 200 s, 1 °C/s) and a

second IR stimulation temperature of 225 °C (pIR$_{225}$ for 200 s, 1 °C/s). The response to a test dose (38 Gy) was measured in

the same way. At the end of each SAR cycle an IR stimulation at 290 °C was used for 100 s to reduce recuperation. An artificial

beta dose of 75 Gy was given to the museum specimens and considered as the natural signal. Eight regenerative dose points

up to 2250 Gy were used to construct the dose response curve, including zero-dose and repeated dose points, allowing

measurement of the characteristic saturation dose (D$_0$). Athermal signal loss was estimated following Auclair et al. (2003) in

order to determine sample specific g-values, which were normalised to a measurement delay time of 2 days after irradiation

(g$_{2days}$; Huntley and Lamothe, 2001). Regenerative doses of 75 Gy, test doses of 38 Gy, and delay times of 100, 200, 400, and

800 min were used including a prompt measurement at the beginning and end of the fading measurements. The first 3.6 s of

stimulation minus a late background of the last 35.5 s were used for signal integration. Single-grain measurements were carried

out by loading the 150-212 μm grains into 300 μm diameter hole single-grain discs. One disc (100 grains) was measured for

each subsample. We checked randomly under a microscope whether multiple grains were bedded in one hole, but multi-grain

holes were infrequent (<1 %). The pIR-IRSL$_{225}$ measurement protocol used for the multi-grain aliquots was modified for

single-grain dose response and fading measurements using an IR stimulation time of 5 s. The first 0.12 s of stimulation minus

the background acquired from the last 1.0 s were used for single-grain signal integration.

## 4 Results

### 4.1 Feldspar characteristics and dissolution features

The Ca, Na and K contents measured by XRF were used to determine the composition of each feldspar sample compared to

the pure calcic, sodic and potassic end-members (Table 1). These analyses showed that the ALB1 and ALB2 samples were

high-sodium plagioclases in the oligoclase range with an accessory K component. Sample ALB1 yielded a higher Ca content.

Sample MIC is a microcline K-feldspar with intermediate characteristics and about 25 % Na; diffraction data suggests partial

exsolution.

The dissolved element concentration in the solution after chemical treatments provided insights into the dissolution extent of

the three feldspar samples. Changes in solution chemistry from both experiments are shown in Figure 1. Most of the dissolved

element concentrations remained relatively constant over the two 720 h experiments. In the aqua regia experiment (Fig. 1a-c),

sample ALB1 showed higher leaching of Ca compared to sample ALB2, which directly reflected the higher Ca content in the

crystal structure of sample ALB1 (Table 1). When considering the alkaline and alkaline earth elements, Ca was the more easily

dissolved cation from the two plagioclases, while for sample MIC, Na, Ca and K reached similar concentrations. Sample MIC

also yielded more Si in solution than the plagioclases. Finally, the solutions composition indicated increasing leaching of Al

over time. Dissolution remained incongruent when comparing the mole ratio of Si to Al in the solids and in the solutions (Table

S1).





Oxalic acid treatment resulted in the leaching of Al, Si, and Na from the ALB1 and ALB2 samples, with lesser amounts of Fe
and K passing into solution (Fig. 1d,e). Very low amounts of elements were leached from sample MIC, even after the 720 h
treatment, with dissolved concentrations of all analysed elements remaining close to or under 1 mM/kg (Fig. 1f). The three
feldspars dissolved congruently in oxalic acid at pH 3 based on the Al/Si mole ratios of the solids and solutions (Table S1).
In comparison with the initial element concentrations of the solids (Table 1), for both solutions, concentrations showed only
minor feldspar dissolution after 720 h with <5 % of total Al, Si, Na, and Ca being extracted and passing into solution, while
5-8 % of the Mn and Fe were extracted.

SEM analyses were performed for each sample before and after chemical treatments (Fig. 2, Fig. S1) and revealed that
dissolution was non-uniform. Grains from the reacted samples exhibited areas that ranged from fresh to altered (Fig. 2). The
most obvious dissolution features were etch pits of few micrometres at the feldspar surfaces, although they occurred irregularly,
demonstrating dissolution heterogeneity. Sample ALB1 revealed more dissolution features at the grain surface compared to
that of samples ALB2 and MIC. For ALB1, discrete secondary precipitates were detected in the form of Fe and Mn phases
(Figs. 2a,b). The SEM photomicrograph showed K-rich laminae at the surface of ALB1 (Fig. 2c), which seemed to be
unaffected by the two solutions over 720 h. The same features were observed at the surface of sample ALB2, although to a
lesser extent. Besides the etch pits at the plagioclase surface and secondary precipitates (Figs. 2e,d), no other grain surface
changes were observed. Sample MIC showed almost no surface dissolution features after aqua regia or oxalic acid treatment,
which is consistent with the low extent of dissolution observed for this sample.

### 4.2 Luminescence dating properties

#### 4.2.1 Thermoluminescence (TL)

TL emission spectra were recorded in the wavelength region between ~280 and ~715 nm, giving information about TL
recombination centres (Fig. 3). Except for their absolute intensities, TL emissions looked similar for the two plagioclase
samples (ALB1 and ALB2) with emissions in the near-UV (~340 nm), blue (~410 nm), and green-yellow (~575 nm)
wavelength bands. A tail of the red emission was visible around 710 nm. The detected luminescence emissions were in
agreement with previous observations (e.g., Krbetschek et al., 1997; Riedesel et al., 2021). All emissions occurred in the low
temperature range at ~100 °C, whilst the near-UV and green-yellow emissions showed a dominant peak at ~250 °C. The blue
emission of samples ALB1 and ALB2 revealed a tail to higher temperatures, but their intensities were lower compared to their
dominant peaks in the lower temperature region. In contrast, sample MIC showed a broad emission ranging from ~370 and
~610 nm, which occurred around 100 °C (Fig. 3). Although much dimmer, the red emission tail is also evident in the region
around 710 nm.

TL emissions of the aqua regia- and oxalic acid-treated subsamples are similar compared to those of the unweathered samples,
except for their TL intensities (Fig. 3, Fig. S2). Whilst mean TL intensities (n=3) for sample ALB1 were up to 15 % higher
after oxalic acid treatment, sample ALB2 showed insignificant changes with chemical weathering except for the green-yellow





emission (Fig. 4). The latter was reduced by ~25 % compared to the unweathered TL emission with both aqua regia and oxalic acid treatments. Sample MIC showed no change with feldspar dissolution, showing similar TL intensities for the near-UV, blue, and green-yellow TL emission bands (Fig. 4). Although TL intensity changes were observed for the two plagioclase

samples, between-aliquot-scatter could explain the slightly higher or lower TL intensities after chemical treatment. When comparing the mean TL intensities of the three aliquots with their corresponding 1-sigma standard deviations (Fig. S3), TL emissions are still identical and overlap the unweathered TL curves (Fig. 5). Although the aliquots were mass corrected, the number of luminescent grains may have varied between aliquots, leading to the differences in TL intensities.

### 4.2.2 Infrared stimulated luminescence (IRSL)

#### 4.2.2.1 pIR-IRSL decay and dose response

The multi-grain feldspar samples generally yielded bright $IR_{50}$ and $pIR_{225}$ signals in the near-UV and blue emission wavelength regions ($T_n$ intensities = several $10^4$-$10^5$ counts per 0.7 s) and $IR_{50}$ signals decayed faster than the $pIR_{225}$ signals, which is well known from previous pIR-IRSL dating studies (e.g., Kars et al., 2014). After artificial chemical weathering, signal decay of the $IR_{50}$ and $pIR_{225}$ signals in the UV and blue luminescence emissions exhibited no change with aqua regia or oxalic acid

chemical weathering (Fig. 6, Figs. S4-5). Although a change in IRSL sensitivity was recorded, there was no clear trend of increasing or decreasing signal intensity. Greater variability in IRSL sensitivity was observed after oxalic acid treatment (Figs. 6c,d, Figs. S4-5). As for the TL data, inter-aliquot scatter must also be accounted for since IRSL intensities are dependent on the number of luminescent grains measured on a single aliquot. In agreement with the luminescence signal decay data, the dose response curves of the three samples in the UV and blue emissions showed no change after artificial chemical treatment

(Fig. 6, Figs. S6-7). We measured the dose response curves up to saturation (2250 Gy) to calculate the characteristic saturation dose (i.e., $D_0$, 68% of the dose response curve), after fitting the data with a single saturating exponential function (Fig. 6, Figs. S6-7). Normalised to the initial $D_0$ values, all $D_0$ values were within 10 % of each other, and did not show differences due to feldspar partial dissolution (Fig. 6, Fig. S8).

Single-grain measurements using the $pIR-IRSL_{225}$ protocol showed bright signals for the $IR_{50}$ signals, while measurement of

the $pIR_{225}$ signal was challenging due to dim signals or anomalous dose response curves (i.e., dose response curves that did not exhibit monotonic growth). Only 2 % of grains could be accepted for sample MIC, whilst samples ALB1 and ALB2 yielded an acceptance rate of 47-95 % of 100 grains for the $IR_{50}$ signal (Table S2). As the samples did not have a natural signal, we evaluated the data following a laboratory evaluation and tested whether we could recover the laboratory administered dose (i.e., a dose recovery test). Dose distributions yielded overdispersion values <3 %, except for sample ALB1 which showed a

higher overdispersion after 240 h aqua regia treatment in the UV emission with 8±1 % (Table S2). Mean dose recovery ratios were all within unity indicating insignificant intrinsic scatter.



#### 4.2.2.2 pIR-IRSL anomalous fading

Measurement of anomalous fading rates were carried out using the pIR-IRSL$_{225}$ protocol at a multi-grain level (i.e., Buylaert et al., 2009; Thomsen et al., 2008) in the near-UV and blue wavelength regions. Figure 7 summarises fading rate data obtained
for the IR$_{50}$ and pIR$_{225}$ signals in both detection wavelengths before (time point 0 h) and after chemical treatment (time points 4-720 h). Measurement of fading rates showed variations in g$_{2d}$-values between the three samples and the two luminescence emission bands. The two plagioclase samples yielded higher fading rates compared to the microcline sample. Moreover, signals emitted in the near-UV are generally less stable than the signal measured in the blue wavelength region (Fig. 7), which is in agreement with previous observations (e.g., Thomsen et al., 2008).

Fading rates of the two plagioclases and the microcline showed little change with weathering time. Most of the fading rates remained 1-sigma uncertainty and did not follow a specific trend (Fig. 7). Sample ALB1 exhibited slightly decreasing fading rates following weathering with oxalic acid in both emissions. Given the recovery of fading rates after 240 h in the blue emission, it is likely that fading rates changed slightly as a function of laboratory measurement uncertainty rather than artificial chemical weathering. Sample ALB2 yielded steady state fading rates in the UV for both pIR-IRSL signals within uncertainty.

In the blue emission, fading rates showed a decreasing trend with weathering time using aqua regia after 4 h, although large measurement uncertainties have been observed. Fading rates decreased from very high rates of 47.3±11.5 %/decade (IR$_{50}$) and 10.7±0.6 %/decade (pIR$_{225}$) to 27.4±6.0 %/decade (IR$_{50}$) and 5.8±2.3 %/decade (pIR$_{225}$), respectively (Fig. 7). However, it should be noted that initial fading rates of the untreated subsamples showed similar rates as for the 720 h aqua regia time point of 29.2±7.5 %/decade (IR$_{50}$) and 7.2±1.2 %/decade (pIR$_{225}$). Laboratory fading rates are likely more biased by measurement
uncertainties as indicated by the large 1σ-standard deviations of the three measured aliquots. Larger scatter has also been observed for sample MIC for both signals and luminescence emissions. Initial fading rates are 2.2±0.4 %/decade (IR$_{50}$) and 1.8±0.3 %/decade (pIR$_{225}$) in the UV and 2.2±0.3 %/decade (IR$_{50}$) and 1.8±0.5 %/decade (pIR$_{225}$) in the blue emission. No significant changes have been observed with aqua regia weathering, all weathering time points showed fading rates, which were within 1-sigma error (Fig. 7).

Since luminescence data at a multi-grain level can be affected by averaging effects (e.g., Reimann et al., 2012; Trauerstein et al., 2014), we also studied the fading properties at the single-grain level. Due to dim pIR$_{225}$ signals, single grain data are presented for the IR$_{50}$ signals for the two plagioclase samples only (Fig. 8). Sample MIC has insufficiently signal intensities for robust fading rate determination at the single-grain level. For samples ALB1 and ALB2, although slight changes can be observed at the single grain level, the initial and final fading rates remain within uncertainty. The most striking observation is
an improvement in measurement precision. As a function of weathering time from 0 to 720 h, standard errors got smaller and precision became better (Fig. 8).



## 5 Discussion

Our experiments showed that greater dissolution occurred for minerals in contact with aqua regia than oxalic acid (Fig. 1), as expected given the stronger acidity, higher oxidation potential, and higher temperature of the aqua regia digestion. The

extraction with oxalic acid however allowed for the leaching of higher amounts of Si due to the presence of oxalate, a complex-forming ligand (Shotyk and Nesbitt, 1992). Additionally, SEM analyses revealed heterogeneously distributed etch pits or solution cracks, similar to observations by Gout et al. (1997). Moreover, few secondary precipitates rich in Fe and Mn were observed at grain surfaces (Fig. 2). Given that feldspar dissolution occurred incongruently after aqua regia and that kinetics of Al and Si accumulation in solution was non-linear (Fig. 1), we assume that preferential diffusion-controlled dissolution

occurred (Lasaga, 1984; Yuan et al., 2019). Thus, the dissolution rate slows down with time, which is likely controlled by the ion substitution between $H^+$ ions and $Na^+$, $K^+$, and $Ca^{2+}$ cations, and subsequent Al-O-Si and Si-O-Si bond breaking (e.g., Gout et al., 1997; Oelkers and Schott, 1995). The faster substitution of hydrogen and alkali ions compared to those of Al and Si was also evident in our experiments (Fig. 1). While feldspar partial dissolution led to the breaking of some Si-O-Si bonds, most of the initial feldspar surfaces remain unaltered. Therefore, feldspar dissolution seems to be controlled by selective chemical

reaction at the solid-solution interface and not by uniform diffusion (Berner and Holdren, 1979).

Feldspar minerals show different chemical resistance to environmental changes (e.g., temperature, moisture, fluid composition) with K-feldspars being generally more resistant than plagioclases. For plagioclases, resistance to chemical weathering decreases with the increasing amount of Ca (Goldich, 1938). This pattern has also been shown in our samples, with more dissolution observed for the Ca-rich plagioclase (sample ALB1) over sample ALB2 and finally with the lowest extent of

weathered features for the microcline (sample MIC) (Figs. 1 and 2). The solution chemistry confirmed that the three feldspar samples behaved differently under artificial chemical weathering (Fig. 1), indicating that feldspar dissolution was dependent upon the chemical composition of the minerals (e.g., Shotyk and Nesbitt, 1992). TL emission spectroscopy has shown that the two plagioclase samples were dominated by the green-yellow emission, whilst the UV and blue emissions were less intense. This emission spectrum may be indicative of an ordered mineral structure (Riedesel et al., 2021); however, high fading rates

(i.e., $IR_{50}$, >10%/decade, Fig. 7) in the near-UV and blue emission bands suggests that the two plagioclases might be characterised by a slight degree of disordering (Spooner, 1994). Structural disorder would predispose the minerals to easier bond-breaking and element release (Yuan et al., 2019). Nonetheless, the effect on the luminescence properties is rather minor as no significant change in the TL and IRSL signals (Figs. 5-8) was observed after 720 h of artificial weathering. Although fading rates appear to reduce with oxalic acid treatment at the multi-grain level, this trend is not apparent at the single grain

level (Figs. 7 and 8).

Potential defects associated with luminescence production might be related to $Mn^{2+}$ for the green-yellow emission (Geake et al., 1977) and Al-O-Al bridges for the blue emission (Finch and Klein, 1999; Speit and Lehmann, 1982). Our results show that $Mn^{2+}$ was leached slightly from the crystal structure (Fig. 1). Assuming that the luminescence centre of the dominant green-yellow emission in samples ALB1 and ALB2 can be associated with $Mn^{2+}$ (Geake et al., 1977), the lower TL intensities around



575 nm following chemical weathering might reflect the leaching of Mn from the crystal's surface, which would indicate a change in TL sensitivity after weathering. As the blue emission likely results from Al-O-Al bridges (Finch and Klein, 1999; Speit and Lehmann, 1982), leaching of Al might play a role on the luminescence signal commonly used for dating purposes (e.g., Buylaert et al., 2012). However, based on our observation of negligible changes in the IRSL and TL properties in the blue emission band, it seems that the blue emission is unaffected by the dissolution of Al from the surface. The same pattern

was observed for the UV emission, which may contribute to the luminescence signal measured when using the blue filter pack. Thus, surface controlled mechanisms in feldspar dissolution characterised by leached surfaces and secondary precipitates seem to have only minor effects on luminescence dating properties. As most luminescence properties remain constant and show a very minor change after chemical weathering, we assume that luminescence originates from the entire grain rather than just the surface where chemical weathering takes place (Holdren and Berner, 1979). Given the improved data precision in the

single-grain data, it might also be assumed that artificial weathering led to cleaning of the surfaces and allowing to measure more of the bulk signal. While this is true for the three feldspar samples under study, these properties may differ for less transparent feldspars (i.e., darker colour) (Duller, 1997).

Dissolution rates under natural conditions differ from laboratory rates, and have been found to be orders of magnitudes slower (e.g., Gruber et al., 2014; White and Brantley, 2003; Zhu, 2005). Using aqua regia at higher temperature (40 °C), we used

aggressive hydrolysis conditions; yet we generally brought into solution less than 5 % of the elements, with the exception of iron (8 %). Longer laboratory treatment times or the use of different ligands (e.g., F$^-$; Shotyk and Nesbitt, 1992) may allow natural weathering conditions over $10^6$ year timescales to be approximated. Very high degrees of etching have been reported for some naturally weathered soil feldspars (Berner and Holdren, 1979). However, natural soil solution contains a wide range of organic and inorganic ions and is prone to ionic strength, pH and redox fluctuations (e.g., Zhang and Furman, 2021). Natural

weathering is furthermore difficult to mimic in the laboratory as biotic components will bring further complexities by exuding compounds specifically meant to extract the limiting nutrients from minerals (e.g., Epihov et al., 2021). In this study, we have nevertheless shown that slight to moderate alterations of the mineral surface (creation of etch pits and oxide precipitates) does not affect luminescence properties important for dating sediments over the datable range (i.e., up to $10^5$ time scales, Bartz et al., 2018). Based on our observations, it seems likely that complex feldspar luminescence properties such as variability in

fading rates (Valla et al., 2016), differences in IRSL intensities (Wang and Miao, 2006), or age reversals (Berger et al., 2001) might not be correlated alone to the degree of chemical weathering, and are probably affected by a complex interaction between intrinsic (mineral properties) and extrinsic factors (environmental properties) (Opolot and Finke, 2015).

**6 Summary and Conclusion**

Feldspar luminescence properties are assumed to be resistant to chemical weathering over Quaternary timescales. For the first

time, we have studied the effect of artificial chemical weathering of feldspars on their luminescence dating properties. Only minor changes were observed, even after aggressive chemical attack using aqua regia for one month. The acids attacked the

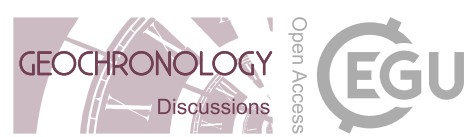

grain surfaces, while the core of the grains remained intact. Whilst the dose response curves were unaffected by the chemical treatments, intensities of TL and IRSL signals may have been slightly modified. However, this result should be taken with caution since other factors may have played a role in causing these differences in signal intensities (e.g., between-aliquot variability). In conclusion, feldspar luminescence properties were mostly unmodified, so that complex luminescence characteristics were likely influenced by other sources of uncertainties rather than the chemical treatments alone.

The experiments conducted in this study were chosen to approximate changes at longer time scales in nature. However, laboratory conditions can differ significantly from natural chemical weathering settings given the complex interplay of physical, biological and chemical factors in the environment, as well as variations in climate, hydrology, pore water chemistry and secondary mineral precipitation. In the future, feldspar minerals of different chemical compositions and their luminescence properties should be characterised in natural weathering profiles (e.g., regolith, soil horizons) and compared to their unweathered parent material in order to shed light on complex luminescence characteristics and, in turn, challenging luminescence chronologies.

**Data availability**

All data generated or analysed during this study are included in this manuscript or its supplementary material.

**Author contribution**

All co-authors contributed to the experimental design, analyses were performed by MB. MB prepared the manuscript with contributions from all co-authors.

**Competing interests**

The authors declare that they have no conflict of interest.

**Acknowledgements**

This work was funded by the Swiss National Science Foundation (SPARK, Grant No: CRSK-2_190905) awarded to GEK. We are grateful to Laetitia Monbaron for laboratory support and ICP-OES analysis. Caroline De Meyer is gratefully acknowledged for helping with the SEM analyses. Olivier Reubi and Thierry Adatte are thanked for their support with the XRF and XRD, respectively, preparation and analyses of the samples.



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





**Table 1: Bulk chemical composition of samples ALB1, ALB2, and MIC determined by X-ray fluorescence (XRF) and expressed as percent oxide. The Ca, Na and K contents were used to determine the composition of each feldspar sample compared to the pure calcic (An), sodic (Ab) and potassic (Or) end-members.**

| Sample ID | Feldspar composition (%) | | | XRF (wt%) | | | | | | | | |
|---|---|---|---|---|---|---|---|---|---|---|---|---|
| | An | Ab | Or | $SiO_2$ | $TiO_2$ | $Al_2O_3$ | $Fe_2O_3$ | MnO | MgO | CaO | $Na_2O$ | $K_2O$ |
| ALB1 | 24.0 | 71.7 | 4.3 | 61.70 | 0.01 | 23.50 | 0.15 | 0.01 | 0.00 | 5.05 | 8.33 | 0.77 |
| ALB2 | 13.8 | 81.6 | 4.6 | 64.71 | 0.01 | 21.70 | 0.13 | 0.01 | 0.00 | 2.93 | 9.55 | 0.82 |
| MIC | 0.4 | 26.3 | 73.3 | 65.61 | 0.01 | 18.85 | 0.10 | 0.01 | 0.00 | 0.09 | 2.94 | 12.46 |








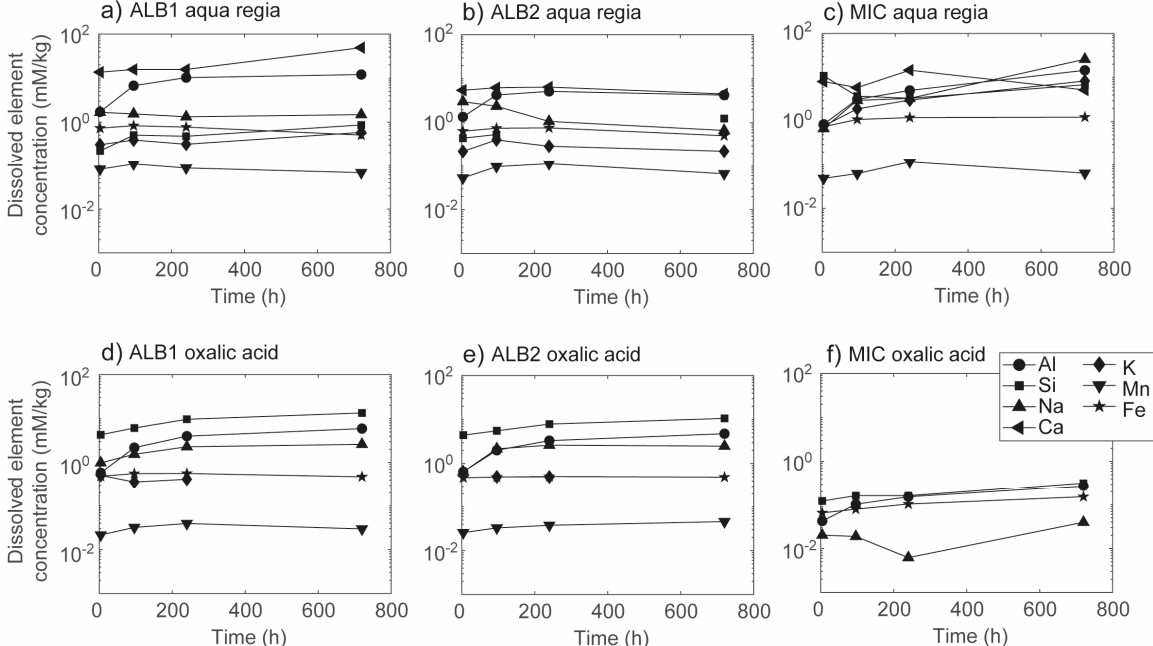

**Figure 1: Extracted elements from samples ALB1, ALB2, and MIC using aqua regia (a, b, and c) as well as oxalic acid (d, e, and f) solutions in the time intervals 4-720 h.**



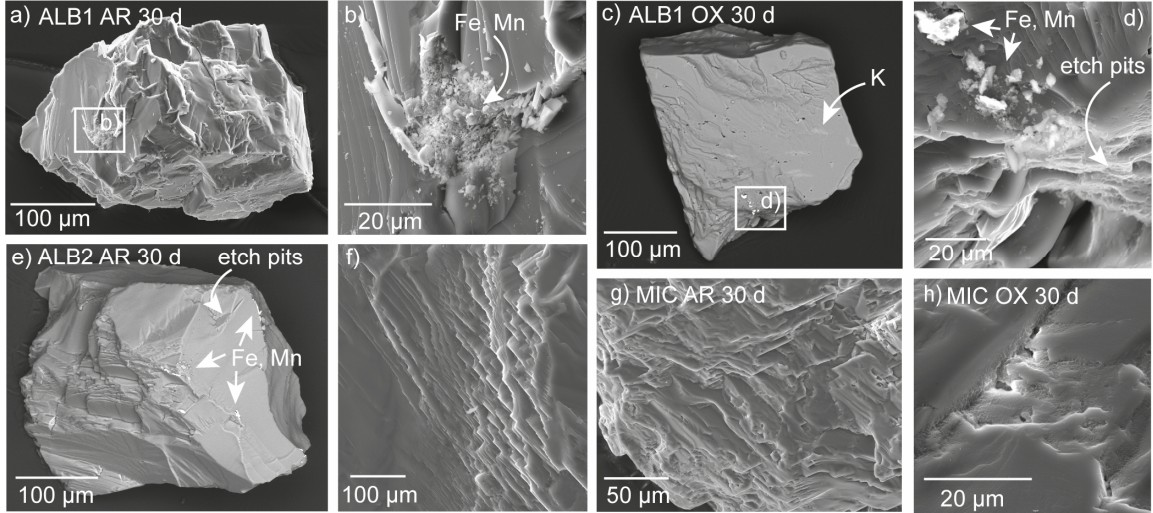


**Figure 2: Scanning electron microscope (SEM) images after chemical treatment using aqua regia (AR) and oxalic acid (OX) for 720 h of samples a-d) ALB1, e-f) ALB2, and g-h) MIC. While image c) was detected with a BSE detector, all SEM images were detected with a SE detector using 20 kV HV and ~21 mm WD.**


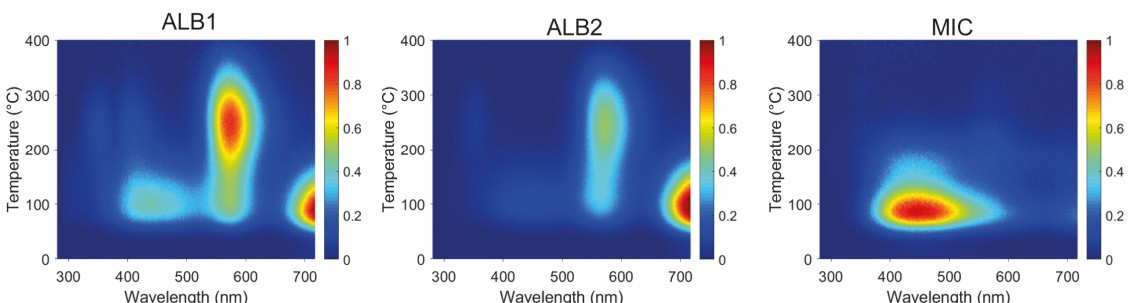


**Figure 3: Thermoluminescence (TL) emission spectra recorded up to 400 °C between ~280 and ~715 nm for samples ALB1, ALB2 and MIC (unweathered). TL intensities are normalised for aliquot mass, dose (270 Gy), and electron multiplying gain to the highest TL intensity. Spectra are corrected for spectrometer efficiency.**




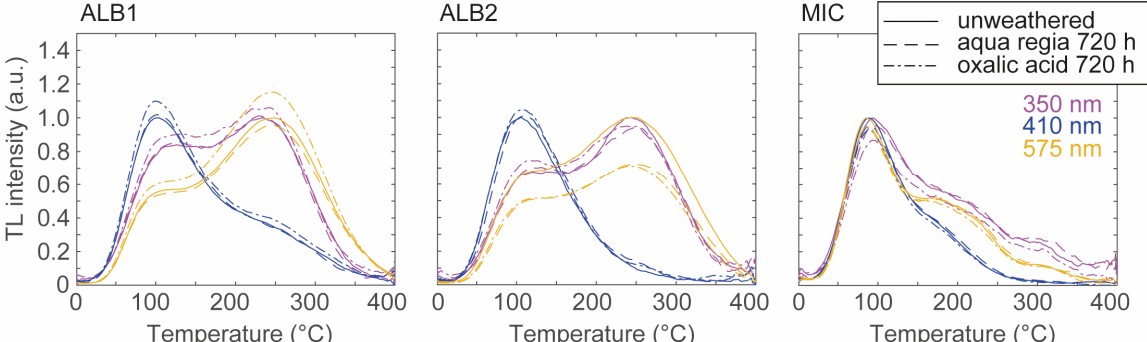

**Figure 4: Thermoluminescence (TL) curves extracted from TL emission spectra for different wavelengths (near-UV, blue, and green-yellow) for the unweathered specimens and the chemically treated samples for 720 h using aqua regia (dashed line) and oxalic acid (dashed-dotted line). The aqua regia and oxalic acid spectra were normalised to the unweathered sample (solid line) for comparison. All curves show the mean values measured from three aliquots.**

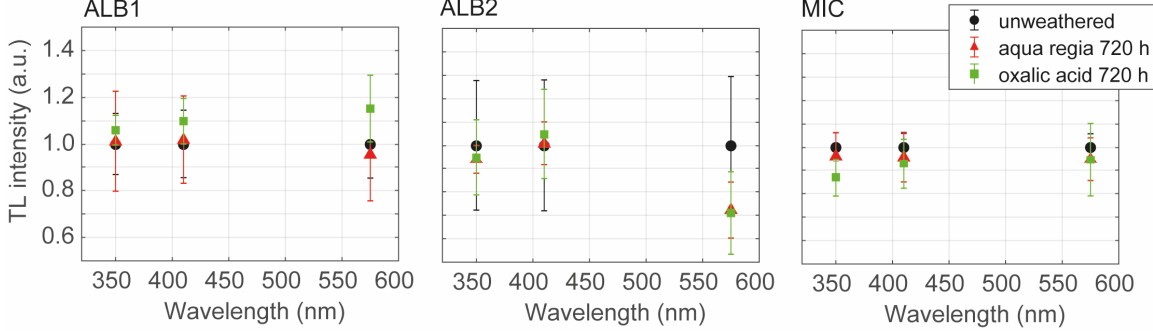

**Figure 5: Thermoluminescence (TL) dominant peak positions extracted from TL emission spectra for different wavelengths (near-UV, blue, and green-yellow) for the unweathered (black circles) and the chemically treated samples for 720 h using aqua regia (red triangles) and oxalic acid (green squares). Aqua regia and oxalic acid spectra were normalised to the unweathered sample. All curves show the mean values measured from three aliquots and associated 1-sigma standard deviation. Dominant peak positions: near-UV = 100 °C (MIC) and 250 °C (ALB1, ALB2); blue = 100 °C (all samples); green-yellow = 100 °C (MIC) and 250 °C (ALB1, ALB2).**





**Figure 6: Exemplary pIR-IRSL signal decay, dose response (inset) (a-d) and characteristic saturation dose ($D_0$; e-f) data of sample ALB1 measured of the blue emission before (0 h) and after aqua regia and oxalic acid treatment (4-720 h). IRSL signals are presented as the mean of three multi-grain aliquots and normalised to the initial (unweathered, 0 h) $IR_{50}$ and $pIR_{225}$ signals. The single saturating exponential function was used to fit the regenerative dose points in the dose response curves. Fitting was performed through three measured aliquots, and normalised to the highest IRSL intensity. $D_0$ values are normalised to the initial $D_0$ values of the $IR_{50}$ and $pIR_{225}$ signals.**



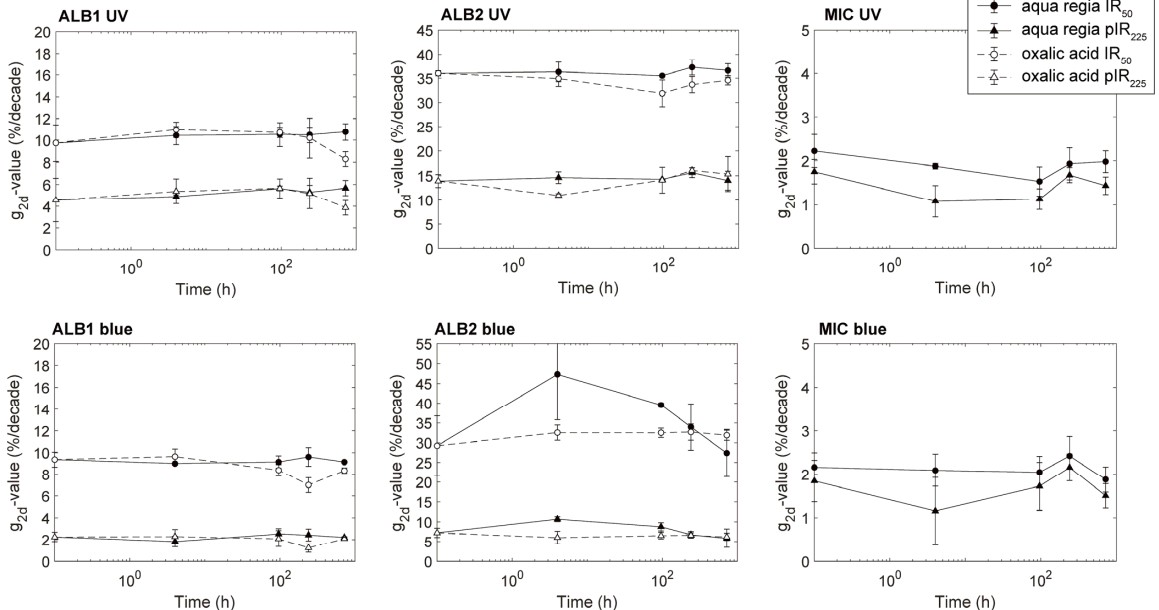

**Figure 7: Multi-grain fading data of the IR$_{50}$ (circle) and pIR$_{225}$ (triangle) signals treated with aqua regia (filled) and oxalic acid (unfilled).**



**Figure 8: Single-grain fading data of the IR$_{50}$ signals illustrated as Abanico plots (Dietze et al., 2016). Fading rates are normalised to 2 days (g$_{2d}$; Huntley and Lamothe, 2001) as a function of weathering time from 0 h (unweathered) to 720 h for the near-UV and blue wavelength regions and two acid solutions (aqua regia and oxalic acid). The mean is centred in the Abanico plots.**