# Peer review of "Potential impacts of chemical weathering on feldspar luminescence dating properties"

_Geochronology, 2022_

## Referee Comment (RC1)

**Potential impacts of chemical weathering on feldspar luminescence – dating properties**

*Bartz* et al.

**General comments:**

The authors have artificially weathered three feldspar mineral specimens (2 albite and 1 microcline) using oxalic acid and aqua regia for a month time. Luminescence characteristics of unweathered and weathered (at different time steps like 4, 10 and 30 days) grains were characterized. TL spectra, TL glow curve, IRSL decay curve, IRSL based dose response curve and fading rate were measured to check the change in the luminescence characteristics of these 3 feldspars. Very little change was observed in all the luminescence characteristics. The authors performed this study (artificial weathering) before they embark on the naturally weathered materials.

Although it is difficult to mimic the natural weathering process in laboratory, as is reflected in an unchanging luminescence characteristics, it is an important work. It is structured and presented in a way it can be understood and reproduced. However, I feel the sample selection is not proper because essentially 2 feldspar specimens were considered (albite and microcline) which under-represent the feldspar types. They could have considered 1) geochemical end members (orthoclase and anorthoclase in addition to albite), and 2) order-disorder representing feldspar (like sanidine in addition to microcline). I have some more comments and suggestions before it is accepted for publication.

**Specific comments:**

1. It is said that XRD measurements were done but the results are not shown. As XRD measurements have already been done, the order-disorder parameter could have been calculated and considered as another luminescence parameter. This parameter becomes relevant as the blue emission is hypothesized to be associated with an oxygen ion trapped in between 2 Al ions (Al-O$^{1-}$-Al). Hence monitoring this order-disorder parameter using XRD measurements with the time points will give better understanding.
2. In line with the earlier suggestion, change in the TL intensity and IRSL intensity before and after the artificial weathering i.e., the effect of weathering in controlling the residence time of trapped electron/hole is a direct and important luminescence dating characteristic. In simple terms whether weathering make a sediment younger than actual. This measurement demand that all the experiments should happen in dark environment. So, the reduction (or no change) in laboratory induced luminescence intensity before and after weathering will suffice (only 2 time points). For measurements shown in Figures 4 and 5, as I understood, laboratory irradiation was given after each time point.

**Technical comments:**

1. L.306-311. It is difficult to understand how $Mn^{2+}$ is invoked to explain a small change in TL intensity (575 nm) with different weathering stage. Because this difference is explained away by the aliquot to aliquot variability in L. 219-223 referring to Fig. 5 and Figs. 3S.
2. L.315. Do you mean by U-340 filter, rather than the blue filter pack?
3. L.339. If this line is modified as "Although feldspar luminescence properties are assumed to be resistant to chemical weathering over Quaternary timescales, there are studies attributed weathering for the stratigraphically inconsistent ages. …. "

Thanks and regards, Morthekai | Birbal Sahni Institute of Palaeoscences (BSIP), Lucknow, India.

---

## Author Response (AR1)

**Response to reviewer's comments:**

**We thank both reviewers for reviewing our manuscript and their constructive comments, which clearly improved the manuscript. We revised the manuscript (including figures in the main manuscript and the supplementary material) and added additional information according to the reviewer's comments (in italic). In the following I will explain our changes to the manuscript in detail (in bold).**

**Melanie Bartz (on behalf of the co-authors)**

**Reviewer 1:**

*«However, I feel the sample selection is not proper because essentially 2 feldspar specimens were considered (albite and microcline) which under-represent the feldspar types. They could have considered 1) geochemical end members (orthoclase and anorthoclase in addition to albite), and 2) order-disorder representing feldspar (like sanidine in addition to microcline).»*

**We agree that testing more samples representing a wider range of natural variability is generally a good idea. Our samples were however carefully chosen for our purposes, because most of the luminescence dating studies focus on K-rich and Na-rich feldspars.**

**Regarding the plagioclase samples, we have chosen two Na-feldspars with different Ca-contents in order to discover whether this would influence their luminescence properties during weathering. As presented in the manuscript, this was not the case. We did not include a Ca end-member (anorthite) because these feldspars are less common in weathered soils, and not representative for most of the luminescence dating approaches.**

**Likewise, for the K-feldspar, we have chosen the lower-temperature polymorph (microcline) because it is the most likely to persist in soils undergoing extensive chemical weathering. Sanidine, which forms in felsic volcanic rocks, is very rare in soils and was therefore not chosen in our experiments.**

*«1.        It is said that XRD measurements were done but the results are not shown. As XRD measurements have already been done, the order-disorder parameter could have been calculated and considered as another luminescence parameter. This parameter becomes relevant as the blue emission is hypothesized to be associated with an oxygen ion trapped in between 2 Al ions (Al-O1- -Al). Hence monitoring this order-disorder parameter using XRD measurements with the time points will give better understanding.»*

**We agree that order-disorder structure of feldspars is an important characteristic of luminescence emissions, which becomes especially true for the Al-O‾Al (i.e., blue emission). However, we think that a detailed discussion about the XRD data will not give much new value to the manuscript. Additional XRD experiments on the treated albite samples after 240 h did not show any differences except for intensity (Fig. R1).**

[Figure]

*Figure R1: XRD results of samples ALB1 and ALB2 of untreated and treated (240 h using aqua regia and oxalic acid) subsamples.*

*«2.    In line with the earlier suggestion, change in the TL intensity and IRSL intensity before and after the artificial weathering i.e., the effect of weathering in controlling the residence time of trapped electron/hole is a direct and important luminescence dating characteristic. In simple terms whether weathering make a sediment younger than actual. This measurement demand that all the experiments should happen in dark environment. So, the reduction (or no change) in laboratory induced luminescence intensity before and after weathering will suffice (only 2 time points). For measurements shown in Figures 4 and 5, as I understood, laboratory irradiation was given after each time point.»*

**We agree that the experimental approach the reviewer suggests (i.e. irradiation before chemical treatments) would be very interesting. We considered using this approach but decided that the experiment would be too complicated. Starting the experiments by irradiating the samples prior to chemical treatments would add additional uncertainty in terms of natural dose distribution and the effect of irradiation without preheating on the distribution of anomalous fading, especially as we chose to carry out the chemical treatments at higher temperatures (40 °C for aqua regia), which could further modify the dose distribution of highly unstable charge. It would be interesting to try this experiment in future work.**

**Reviewer 2:**

*«Please reflect on the terminology used: I found the term 'weathering' confusing where used as sample label. You are working on museums specimens that will probably not have been exposed to weathering in nature. So may be use 'treated', 'etched', or other terms that better describe the state of a sample. Please use similar terms throughout the manuscript (and also in figure captions).*

*A related issue: In the discussion, please highlight that your results were obtained on museum specimens. Mineral grains extracted from sediments may respond significantly different to etching due to very variable pre-depositional mechanical and chemical alteration.*

*Parts of the results and discussion section need more precise wording, loosing superfluous words and jargon. A job for the senior author?»*

**We agree that «weathering» is not a proper term for laboratory treatments and we decided on using «treated» throughout the manuscript. In general, we worked on the wording throughout the manuscript.**

*Line 67: Wording – what is 'challenging' about putting an adequate optical filter in the detection pathway?*

**We mentioned the challenge of measuring a pure luminescence emission as the blue filter pack still transmits a small part of the UV wavelength region. Small changes in luminescence properties with chemical treatment in the UV emission might also have an effect in the blue luminescence emission usually used in luminescence dating applications. We added information to the manuscript (lines 64ff).**

*Line 73: '...well as changes' change to '...well as to changes'*

**We worked on the grammar/wording throughout the manuscript.**

*Line 104: '(unweathered)'? would '(untreated)' be more adequate? Please print units with all digits: 0 h, 4 h, 96 h, 240 h, and 720 h.*

**Done. We use the terms «treated» or «untreated» in the revised manuscript.**

*Lines 109 & 110: wording – change: 'hydrolysis conditions, which allow it to efficiently leach transition metals and trace elements from the surface of minerals without destructuring the silicate' to 'hydrolysis conditions and efficiently leaches transition metals and trace elements from the mineral surface without destroying the silicate'*

**Done. We worked on the grammar/wording throughout the manuscript.**

*Line 105 versus line 115 – please clarify: was shaking applied in both experiments over the full period?*

**Yes, shaking was applied in both experiments. We made the experimental conditions clear in the methods section of the manuscript (lines 105ff).**

*Line 120: wording – 'weathering time point' change to 'experiment duration' or similar*

**We agree that «weathering time point» might not be the most appropriate term and we changed it to «treatment duration» throughout the manuscript.**

*Line 123: Unclear – 'Sc was used as an internal standard'. Have values been normalized to Sc concentration?*

**The Sc standard was used for investigating the stability of the ICP-OES measurements. A normalization approach was not applied as the Sc readings were constant, which means that the analysis was stable and that normalization would not change the results. We added information to the manuscript (lines 122ff).**

*Line 139: Change 'were used' to 'was used', Line 143: Change 'were weighted' to 'was weighted', Line 168: Change 'were used' to 'was used'*

**We worked on the grammar throughout the manuscript.**

*Line 149: Unclear: 'background noise' – given your measurement setup I expect that the majority of background signal was due to black body radiation and not noise. Did you check the reproducibility of the setup?*

**We agree that the background signal was very minor compared to that of the black body radiation in the high temperature region (>350 °C). However, the signal-to-noise ratio was lower in the low temperature region, which corresponds to the main luminescence emissions for our samples and correction/subtraction of the background signal was necessary. We checked the reproducibility of the spectra between aliquots (n=3) and explored only minor differences in intensity. We added additional information to the manuscript (lines 149ff).**

*Line 160: Please print units with all digits., Line 204: Please print units with all digits.*

**Done.**

*Line 171: 'feldspar sample compared to the pure' change to 'feldspar sample and given as percentage of the pure'*

**Done.**

*Line 173: 'higher Ca content' needs a comparator - compared to?*

**Sample ALB1 was compared to sample ALB2 as both samples are albites. We made it clear in the manuscript.**

*Line 198: 'The same' better 'Similar'*

**We worked on the wording throughout the manuscript.**

*Line 215: wording – While the shape of the TL emissions is unaltered even after …. the intensities of the TL emissions increased….*

**We changed those sentences (lines 215ff).**

*Lines 219 - 223: wording – is it 'no significant change can be detected due to inter-aliquot variability?*

**This is correct. Inter-aliquot variability is more evident rather than «real» changes in TL intensities after chemical treatment.**

*Chapter 4.2.2 – please reword. Shorten and focus using more precise terminology and descriptions. Some suggestions here:*

*Similar lines 230 - 233 – is the scatter significant? If not, refrain from speculating.*

*Line 235: 'up to saturation' - be more precise: The highest applied dose was 2250 Gy – the dose response (S&) is still growing…*

*Line 236: unclear – delete: 'Normalised to the initial D0 values,'*

*Line 241: change 'could be' to 'were'*

**Thank you for your suggestions of modification, we worked on the wording of the whole manuscript to be more precise and to avoid speculative wording.**

*Discussion lines 280-290. Is this a good analogue for chemical weathering in, for example, a soil?*

**It is comparable to weathering processes of specific minerals. Whether those processes can be good analogues in natural environments depends on the complex interplay between intrinsic (mineral properties) and extrinsic factors (environmental properties).**

*Line 315: unclear – 'when using the blue filter pack.' Is it 'the 410 nm emission.'?*

**Yes, it is the 410 nm emission, we made it clear in the manuscript (line 314).**

*Figure 1:*

*I find the semi-logarithmic graph design not very clear. May I suggest plotting the data as percentage change on a linear scale? You can show all results in the same graph by normalizing the concentrations to the initial vale. This should reveal potential trends more easily. In addition to the different symbols, you may also want to use a colour scheme.*

**Following the reviewer's comment, we modified Figure 1 (Fig. R2). However, showing a percentage change relative to the first measurement (no treatment) can indicate a significant change even if the chemical change is minor. This becomes especially true for Mn and Fe where only small amounts of element concentrations were measured in the original minerals. Therefore, we would like to present the original Figure 1 showing dissolved element concentrations of treatment durations 4-720 h. In addition, plotting all data in one graph would likely overload the figure and make it difficult to follow due to the amount of data points and associated symbols/colours. Therefore, we kept the number of figures.**

[Figure]

*Figure R2: Dissolved element concentrations subtracted from the initial concentrations of the untreated minerals and normalised to the initial concentrations.*

*Figure 5, Caption: Unclear – 'All curves'? is it 'All symbols'?*

**Yes, this should be «all symbols», we revised the caption.**

---

## Author Response (AR2)

Dear Prof. Feathers,

Thank you very much for reviewing our revised manuscript. Your comments improved it further. In the following, our modifications are explained (bold) according to your comments (italic).

Best regards,

Melanie Bartz (on behalf of the co-authors)

*This is a good and interesting paper, well-written and the responses to the reviewers were generally good.*

**Thank you very much.**

*I have a few comments. If these are addressed, then I think the paper should be published:*

*1. In lines 34-35, the authors cite previously studies observing that chemical weathering affects luminescence properties, yet the thrust of this paper is to say that weathering has little effect. Can you explain these different conclusions?*

**The results can differ due to different sample and experimental conditions:**
- **Whilst previous luminescence studies (Berger et al., 2001; Wang and Miao, 2006; Valla et al., 2016) focused on natural samples, we used samples of specific feldspar types (i.e., similar weathering conditions throughout the bulk sample), whereas natural sediment samples are characterized by mixtures of different feldspars (i.e., feldspars with different strength to chemical weathering).**
- **Laboratory conditions differ from natural conditions (e.g., organic and inorganic ions, ionic strength, pH and redox fluctuations). Please, see discussion lines 323ff.**
- **Natural feldspar samples are likely prone of various erosional-depositional cycles and thus different weathering conditions.**

**In conclusion, it remains difficult to connect the cited studies with our observations. We added explanations to the conclusion.**

*2. Line 117 -- Can "complex" be used as a verb? Maybe reword.*

**Yes, it can be used as a verb in chemistry (like "chelate"). We prefer to keep "complex".**

*3. Lines 212-214 -- Your spectra results show nothing in the high temperature region of MIC, yet the glow curves clearly show a luminescence signal at high temperatures. Can this be explained?*

**This is an effect of data normalization as the data were normalized in two different ways: In figure 3, the data were normalized to the highest TL intensity coming from ca. 450 nm. In**

contrast, figure 4 shows a normalization to the curve itself (e.g. for 350 nm: normalization to the highest TL intensity in the 350 nm region). We added these information to the figure captions.

*4. I thought Reviewer 1's comments on XRD were good. I wonder if you could incorporate your reply into the supplementary material*

We agree and added a figure to the supplementary material showing the XRD results of the albite samples comparing untreated and treated material. Additionally, we added information to the manuscript (lines 178-179).